# FUND-RELATED GRAPH REPRESENTATION FOR MARGINAL EFFECTIVENESS IN MULTI-FACTORS QUANTITATIVE STRATEGY

## ABSTRACT

With increasing research attention in the quantitative trading community on multi-factors machine learning strategies, how to obtain higher-dimensional and effective features from finance market has become an important research topic in both academia and industry area. In general, the effectiveness of new data, new factors, and new information depends not only on the strength of their individual effects but also on the marginal increment they bring relative to existing factors. In this paper, our research focuses on how to construct new factors from the relational graph data. We construct six capital flow similarity graphs from the frequency of joint occurrences of the inflows or outflows of the net fund between stocks within the same period. Moreover, three composite multi-graphes from the six basics are built to exploit the capital flow similarities. Experiments demonstrate the marginal improvement contributing to these proposed graphs. Learned by the multi-factor XGBoost model, the new dataset integrating with representations of the fund-related graphs exceeds the baseline multi-factor model in the Information Coefficient(IC), TOP group excess returns, long-short returns, and index-enhanced portfolio returns in A-share market.

## 1 INTRODUCTION

Recently, with the outstanding performance of deep learning in CV He et al. (2016),Goodfellow et al. (2014),Song et al. (2020),Goodfellow et al. (2016) and NLP Hu et al. (2021),Vaswani et al. (2017),Brown et al. (2020),Liu et al. (2023) area, how to apply machine learning format in the quantitative strategy gains increasing interests in the finance. Machine learning methods have gained increasing attention in the field of quantitative finance. Since the advent of Alpha101, more and more researchers have explored the leverage of technology-driven multi-factor strategies for the representation learning and stock predicting from financial time series data. All led to the dramatic development of a range of multi-factor representation models. Moreover, advancements in storage, computing power, and model methodologies have provided a solid foundation for accumulating effective factors on a large scale in quantitative strategy research. The research ranges from factor-discovering, and representation learning, to model-building. In this paper, our work mainly focuses on the factor-discovering area. Currently, the number of existing factors involved in machine learning quant methods ranges from hundreds to thousands, ushering in a high-dimensional era in asset pricing and factor selection.

In the format of multi-factor quantitative stock selection, the role of factors, whether selected or discovered through technical means, has gradually shifted from signaling Fama & French (1993) to diverse feature representations. Consequently, the evaluation of factor effectiveness has transitioned from single factor efficacy to marginal increments relative to a set of proven effective existing factors. Especially after the widespread application of machine learning and deep learning models in the aspect of factor synthesis, the marginal contributions that a single factor can make under linear dimensionality in terms of low correlation and overcoming multi-collinearity may not be substantial in nonlinear modeling scenarios. Therefore, it is crucial to emphasize how new factors can provide incremental improvements relative to the existing factor pool. in the current research and practice. The initial goal of our research is consistent with this point.

As said in the Arnott et al. (2018), the capital markets reflect the actions of people, which may be influenced by others' actions and by the findings of past research. As a result, one of the crucial challenges becomes how to extract exact data of market participants. Currently, the application of relationship-based graph data in factor mining scenarios has attracted increasing attention in research Saha et al. (2022). Existing academic literature has demonstrated significant lead-lag relationships among returns based on relationship dimensions such as supply chains, analyst coverage, shared news coverage, and industries. In this paper, our research focus on the frequency of joint occurrences of the inflows or outflows of the net fund between stocks within the same period. Momentum transmission factors have been constructed based on these types of information. In the context of high-dimensional stock factors, where factors are used as sample features for asset pricing, relationship-based data can provide incremental information for asset pricing through factor derivation and joint training, leveraging integrated models and graph neural networks in machine learning and deep learning. This can lead to excess returns in quantitative strategies.

Building upon previous studies on relationship-based graphs in factor modeling scenarios, this study utilizes data related to capital flow directions within the market to construct a relationship matrix that represents correlated features among companies. We construct six basic graphs from fund-related data to exploit the representation of capital flow similarities, and three composite multi-graphes are derived for better performance. In detail, we ultimately establish six matrices that represent the similarity of capital flows among stocks through analysis and comparison. Apart from the proposed graphs, feature engineering is conducted on the existing proven efficient factor library, and the integration of various relationship graphs through factor derivation is incorporated during model training and testing process. The results exhibit increased IC values and enhancements in portfolio construction. On the basis of the stock factor library, the capital flow similarity matrix is used for the enhancement of the model training process by training the traditional machine learning model in batch derivative. The experimental results prove that the addition of capital flow similarity brings an enhanced effect on the revenue prediction.

## 2 RELATED WORKS

### 2.1 MACHINE LEARNING IN FINANCE

As machine learning continues to evolve, an increasing number of models are being introduced for financial quantificationGiglio et al. (2022), Jiang et al. (2020). Gu et al. (2020) explored the integration of machine learning techniques into empirical asset pricing. The study delved into how machine learning algorithms can yield significant financial benefits for investors through their capacity to capture nonlinear interactions among predictors. In the pursuit of constructing superior factor models, there has been a shift in recent years from traditional linear models to the adoption of more adaptable and data-centric nonlinear machine learning models. Gu et al. (2021) represented factor exposures as a dynamic nonlinear function of covariates using autoencoder neural networks, a dimension reduction technique. Duan et al. (2022) treated factors as latent random variables in the Variational Autoencoder (VAE) and inherently possessed stochasticity. The proposed model Factor-VAE was capable of modeling noisy data and providing estimates for stock risk. Meanwhile, numerous machine learning algorithms are employed to unearth additional factorsKakushadze (2016) Zhang et al. (2020). In this paper, our research mainly focus on the features required for machine learning models.

### 2.2 RATIONAL DATA FOR THE STOCK PREDICTING

In reality, businesses do not exist in isolation; the relationships between companies form a complex network. Existing literature has identified various interconnections used to explain the future expected returns of assets, including industry relationships, geographical proximity, supply chain connections, stocks held by the same mutual funds, technological similarity between companies, shared analyst coverage, and common news disclosures. Moskowitz and GribblattMoskowitz & Grinblatt (1999) were among the first to propose the presence of strong lead-lag effects in the expected returns of stocks within the same industry. Parsons et al.Parsons et al. (2020) linked companies headquartered in the same region based on geographical proximity and, by constructing geographical momentum portfolios, found that the lead-lag relationships between stocks of companies located in the same city exhibit significant predictive power for their future returns. Cohen and Frazzini Co-

hen & Frazzini (2008) established associations between different companies based on their supply chain relationships and discovered that when the stock returns of one company exhibit volatility, the stock returns of companies interconnected through the supply chain are also impacted, resulting in correlated fluctuations. Menzly and Ozbas Menzly & Ozbas (2010) characterized the lead-lag effects between company supply chains as having explanatory power and predictive capabilities for the expected returns of a company's stock. Anton and Polk Anton & Polk (2014) noted that stocks held by the same mutual funds exhibit noticeable correlations in their returns. This ownership-based relationship among stocks can lead to interdependencies in the price fluctuations of these stocks, potentially even triggering contagion during periods of panic. Lee et al. Lee et al. (2019) conducted an analysis of the mutual predictability of returns among technologically related companies. Typically, companies engaged in patent research and development tend to influence each other, as they often face similar economic shocks within the same technological domain. This can lead to collaborative development efforts, the presence of supply chain connections, or the utilization of common production inputs among these companies. Ali and Hirshleifer Ali & Hirshleifer (2020), in their study of the U.S. market, highlighted that companies associated based on shared analyst coverage exhibit more pronounced mutual influence on their future returns. The authors argue that the number of common analysts can serve as an indicator of the degree of association between these companies. Chen et al. Chen et al. (2021) and others argue that relationships defined by the joint mentions in news media events can better reflect the intrinsic and comprehensive connections between companies. This approach allows for a more accurate characterization of the impact of momentum spillover effects among stock assets.

The aforementioned research findings demonstrate the crucial cross-temporal predictive role of the interrelationships between companies in explaining the future expected returns of assets and the volatility of the securities market.

## 3 METHODOLOGY

### 3.1 DEFINITION OF CAPITAL FLOW SIMILARITY GRAPHS

It is known to all that the capital flow similarity between stocks captures the similarity of trading behaviors that drive stock price movements at the fund level. As a result, it reflects the common expectations of fund participants on the confidence of the stock prices. In other words, stocks with stronger consistency in fund inflows and outflows potentially exhibit stronger correlation.

According to the prior research on modeling approaches such as supply chain relationships, analyst coverage, and shared news coverage, we describe the inter-stock relationships based on common fund flow. The correlation is defined by the adjacency matrix of the involved stocks. Consequently, six capital flow similarity graphs are constructed. The capital flow data is derived from the secondary high-frequency market data and differentiates fund types, transaction types, and capital flows based on information such as order volume, transaction volume, and transaction type.

As a common opinion, transaction types in financial markets can be categorized as active buying or active selling. Based on the order submission time of the buyer and the seller, the earlier order submission is considered the passive party, whereas the later order submission is considered the active party. The definitions of the two categories are shown as follows:

a. Active buying refers to the state when the buyer actively executes a transaction at the seller's ask price, indicating the buyer's willingness to purchase a security or stock at a higher price. This behavior reflects a positive attitude towards the market, indicating optimism. Therefore, active buying shows the investor's evaluation of the market, believing that the security has good growth potential or the potential to generate short-term profits.

b. Active selling refers to the state that when the seller actively executes a transaction at the buyer's bid price, indicating the seller's willingness to sell a security at a lower price. This may imply a lack of confidence in the future performance of the security. Active selling may occur when investors anticipate market or security declines or when investors want to realize profits.

Additionally, based on transaction types and volumes, we can determine fund inflows and fund outflows to assess the direction and scale of capital flows in the financial market. Specifically:

a. Money Flow In (MFI) refers to investors investing capital into a specific financial asset or market, causing its price to rise. This usually indicates optimistic investor sentiment, suggesting that the asset or market has the potential for higher returns.

b. Money Flow Out (MFO) refers to investors withdrawing or selling capital from a specific financial asset or market, leading to a decline in its price. This often indicates cautious or pessimistic investor sentiment, suggesting possible risks or lower returns.

Furthermore, the daily Money Flow Net (MFN) is calculated as the difference between the total value of active buying transactions and the total value of active selling transactions, that is:

$$MFN = MFI - MFO. \tag{1}$$

Here, $MFI$ represents the total value of active buying transactions, while $MFO$ represents the total value of active selling transactions. Therefore, when $MFN > 0$, it is considered a Money Flow Net Inflow ($MFNI$), and when $MFN < 0$, it is considered a Money Flow Net Outflow ($MFNO$). Ultimately, based on the net inflows and outflows of funds between stocks on a given day, we can calculate the consistency of capital linkage between pairs of stocks.

## 3.2 Six Prior Capital Flow Similarity Graphs

As mentioned in the prior part, our research primarily focuses on relational data in the financial market related to fund flows, such as customer order sizes and the order sequence. Regarding the magnitude of fund flows, we clearly classify customer orders into four categories based on the transaction amounts: small orders (retail, $< 40,000$ RMB), medium orders (intermediate, $40,000 - 200,000$ RMB), large orders (institutional, $200,000 - 1,000,000$ RMB), and extra-large orders (institutional, $> 1,000,000$ RMB). Furthermore, based on the order submission time of the buyer and seller, transaction types are classified as passive buying or active buying. Specifically, transactions with a price greater than or equal to the asking price are considered active buying, while transactions with a price less than or equal to the bid price are considered active selling.

Based on the related content of fund flows, combining the characteristics of the stock market and the entities and operations involved in fund flows, six capital flow similarity graph data are constructed as follows:

For companies $i$ and $j$, the capital net flow on day $t$ is represented by $MFN_{it}$ and $MFN_{jt}$. The definition of the daily co-inflow association matrix is shown as follows,

$$ei_{ijt} = \begin{cases} 1, & MFN_{it} > 0 \quad and \quad MFN_{jt} > 0, \\ 0, & others. \end{cases} \tag{2}$$

Given that there are approximately 22 trading days in a month, the monthly co-inflow association matrix for the companies is defined as:

$$EI_{ij} = \sum_{t_{start}}^{t_{end}} ei_{ijt} \tag{3}$$

Here, $t_{start}$ represents the first trading day of the month, and $t_{end}$ represents the last trading day. $EI_{ij}$ represents the element value of the adjacency matrix for capital co-inflow (CashCoIn). Similarly, the daily and monthly co-outflow association matrix for the companies is defined as:

$$eo_{ijt} = \begin{cases} 1, & MFN_{it} < 0 \quad and \quad MFN_{jt} < 0, \\ 0, & others. \end{cases} \tag{4}$$

$$EO_{ij} = \sum_{t_{start}}^{t_{end}} eo_{ijt} \tag{5}$$

The $t_{start}$ represents the first trading day of the month, and $t_{end}$ represents the last trading day. $EO_{ij}$ represents the element value of the adjacency matrix for capital co-outflow (CashCoOut).

Furthermore, MFN is combined with our proposed four customers' transaction amounts and divided into categories: MFNsmall (small orders, $< 40,000$ RMB), MFNmed (medium orders, $40,000 - 200,000$ RMB), MFNlarge (large orders, $200,000 - 1,000,000$ RMB), and MFNexlarge (extra-large orders, $> 1,000,000$ RMB). Therefore:

$$MFNMedSmall = MFNsmall + MFNmed, \tag{6}$$

$$MFNExNLarge = MFNlarge + MFNexlarge. \tag{7}$$

Here, MFI represents the total value of active buying transactions, while MFO represents the total value of active selling transactions. MFNsmall, MFNmed, MFNlarge, and MFNexlarge represent the differences between the active buying and selling transaction amounts for small, medium, large, and extra-large orders, respectively.

Similarly, according to MFNMedSmall and MFNExNLarge, small and medium simultaneous capital inflow matrix (MedSmallCoInAct), small and medium single capital outflow matrix (MedSmall-CoOutAct), large single capital inflow matrix (ExNLargeCoInAct), and large single capital outflow matrix (ExNLargeCoOutAct) are constructed. Finally, our proposed six fund-related graph factors are orgnized as follows:

**CashCoIn, CashOut, MedSmallCoInAct, MedSmallCoOutAct, ExNLargeCoInAct, ExN-LargeCoOutAct**.

To maintain sparsity in the graph data and preserve the strongest relationships, in practice, only the top 1% of strongest connections for each company in each row of the adjacency matrix are retained as neighbors in the graph. The processing will preserve the strongest relationship. Taking the capital co-inflow graph as an example, the detailed construction steps are as follows:

(1) Count the number of trading days in which two stocks have simultaneous capital net inflows over the past 22 trading days.

(2) Build an adjacency matrix for all the stocks in the market, with the assumption that there are correlated relationships between pairs of stocks in terms of capital net inflow. Fill in the element of the adjacency matrix at the corresponding positions of the two stocks with the number of trading days with simultaneous capital net inflows.

(3) Repeat the above steps for all stocks. The resulting values in the adjacency matrix represent the number of trading days with simultaneous capital net inflows for each pair of stocks.

(4) Keep only the top 1% of the strongest connections for each stock in each row, considering only the strongest edges in the network.

Based on capital flow similarity graph, the final relation matrix Mc, representing the capital co-inflow association between stocks, can be constructed as follows:

$$Mc = \begin{bmatrix} Mc_{11} & Mc_{12} & ... & Mc_{1n} \\ Mc_{21} & Mc_{22} & ... & Mc_{2n} \\ ... & ... & ... & ... \\ Mc_{n1} & Mc_{n2} & ... & Mc_{nn} \end{bmatrix} \tag{8}$$

Where $Mc_{ij}$ represents the capital flow co-inflow similarity between the $i-th$ and $j-th$ stocks, representing the number of trading days with simultaneous capital net inflows between the two stocks. Similarly, six maps and their corresponding adjacency matrix matrices are constructed according to different capital types, inflow and outflow conditions.

### 3.3 DERIVATION OF FACTORS BASED ON CAPITAL FLOW SIMILARITY

Using the capital flow similarity matrices mentioned earlier, we further process the factors by creating a weight matrix. To obtain it, each element in the similarity matrix is divided by the sum of its respective row.

Let's consider a specific factor vector $F = \begin{bmatrix} F_1 \\ F_2 \\ ... \\ F_n \end{bmatrix}$, which represents the factor values corresponding to each stock. This vector is of size $n1$. Each element $F_i$, represents the factor value for the $i_{th}$

stock. we multiply the weight matrix ($Mw$) with the original factor vector $F$ to calculate the derived factor vector $F_c$ :

$$F_c = Mw \times F. \tag{9}$$

Among them, $F_c$ is an $n \times 1$ vector (referred to as the derived factor) that integrates the derived factor values of stocks related to capital flow. For a specific stock $i$, if its original factor value is $F_i$, then the derived factor value is $F_{ci} = \sum_{j=1}^{n} Mw_{ij} \times F_i$. To be noticed, the factor value of the derivative factor represents the result of the weighted sum of the factor value of the individual stock related on the capital flow similarity network according to the correlation degree between them.

### 3.4 COMBINATION OF MULTIPLE DERIVED FACTORS

In the model training process, for a given set of original sample features $X = \left[F_1^T, F_1^T, ..., F_M^T\right]$ , the derived feature dimensions $Xw = Mw \times X = \left[Mw \times F_1^T, Mw \times F_1^T, ..., Mw \times F_M^T\right]$ are obtained using the factor graph derivation method. Furthermore, the original sample features and derived feature dimensions are concatenated to form new sample features $\bar{X} = [X, Xw] = \left[F_1^T, F_1^T, ..., F_M^T, Mw \times F_1^T, Mw \times F_1^T, ..., Mw \times F_M^T\right]$.

For a stock $i$ on a given trading day, its corresponding sample $X_i$ represents the stock's own information as features in the sample space, while $\overline{X_i^{\sim}}$ represents the integration of information from stocks related to capital flow similarity in the corresponding relationship network of the factor graph. Thus, in the model training and prediction part, the $[X, X_c]$ is used to sample features for the purpose of leveraging the information both of the stocks and other related stocks. These factors combine through capital flow similarity to jointly predict i stock's returns. Here, we provide some arrangements for Multi-Graphs in Table 1.

During model training, the factor graph integration scheme includes a single graph scheme and a multi-graph scheme: (1)Single Graph: the model $y = g(X)$ trained using original features X serves as the baseline and the incremental improvement of the integrated model $y = g(X) + f(\bar{X})$ compared to the original model is examined. (2)Multi-Graph: The results of training multiple graphs are integrated.

| Multi-Graphs Scheme | Referred Graphs |
|---|---|
| TRI | CashCoIn |
| | ExNLargeCoInAct |
| LRI | ExNLargeCoInAct |
| | ExNLargeCoOutAct |
| TRU | ExNLargeCoInAct |
| | ExNLargeCoOutAct |
| | CashOut |
| | CashCoIn |
| | MedSmallCoInAct |

Table 1: The Scheme of Multi-Graphs is shown in this table. Three different strategies are proposed following the basis of former research.

## 4 EXPERIMENTS

This section evaluates the effectiveness of derived factors based on capital flow similarity in a multi-factor quantitative model using a stock factor library. It compares various schemes, including the standalone application of the factor library, the addition of capital flow similarity graph integration for training, and evaluates the incremental effects of derived factors on the model's performance.

### 4.1 SETTINGS

Our research focuses on the A-share market within the daily data. For the machine learning models used in this study, an XGBoost algorithm is employed, as shown in the Fig. 1. In details, rolling training is conducted every 10 trading days, with each training updating features for the next 10

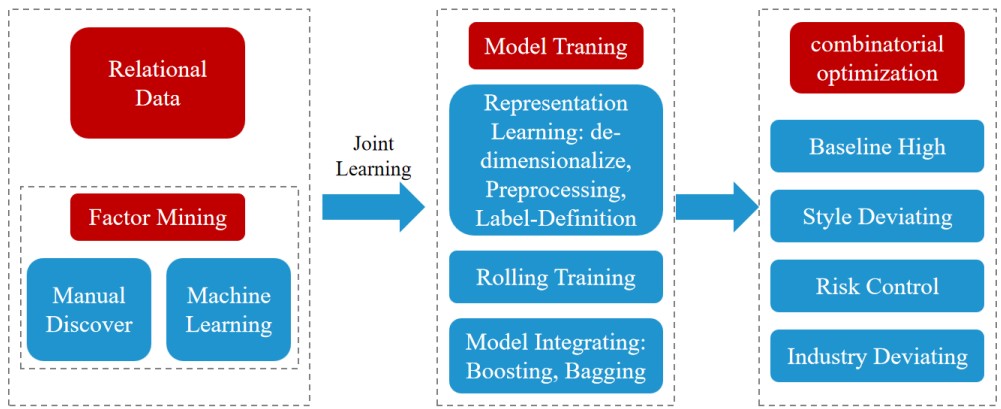

Figure 1: The training Framework for the proposed Relational graph data in finance area.

trading days for stock selection and portfolio construction. All six capital flow similarity graphs, derived from the most recent 22 trading days, is used for factor derivation and feature construction in each training step. The detailed model training parameters are shown in the Table 2.

To evaluate the incremental effects of capital flow similarity graph-derived factors on existing factor libraries, the following parallel control experiments are conducted:

- RawModel: Each training step uses the original factors as individual stock features.
- Integration Graph Factors: Based on the original factors, we perform matrix multiplication using each derived capital flow similarity relation matrix to produce a new derived factor in each step.

This results in each sample come from the double number of factors including features—original factors and the derived factors. As a result, the models trained following the proposed approach utilize both the self-characteristics of individual stocks and the information of stocks related to them fund-related flow similarity to achieve a better prediction of the stock's returns.

| Model | XGBOOST |
|---|---|
| Hyper-Parameters | booster: gbtree; colsample_bytree: 0.7 learning_rate: 0.1; max_depth: 7 min_child_weight: 10; n_estimators: 100 subsample: 0.6 |
| Label | The rate of return calculated by vwap for the next 10 days |
| Pre-processing | Factors:Winsorize shrinks the left and right 5% of the cross-sectional data, normalizing Label: Take the rank quantile Sample dataset: Eliminate samples with a missing value ratio greater than 20% |
| Training Setting | Retrain the model every ten trading days Use samples of 300 trading days each time as the training set Each time use the capital flow similarity map calculated in the last 22 trading days for factor derivation and feature construction |

Table 2: The settings of our experiment is shown in the above table, including hyper-parameters, training settings, labels, and data-processing.

## 4.2 CORRELATION ANALYSIS OF RETURNS FOR STOCKS WITH SIMILAR CAPITAL FLOWS

To assess the consistency of price volatility among stocks that exhibit similar capital flow relationships, we conducted a semi-annual analysis from 2018 to 2021. The analysis focused on the distribution of correlation coefficients for returns among two groups: the entire sample of stocks

in the A-share market, and a subset of stocks that demonstrated similar capital flow relationships within the past 6 months. The results of this analysis are presented in Figure 2.

| | mean | std | min | 10% | 20% | 30% | 40% | 50% | 60% | 70% | 80% | 90% | max |
|---|---|---|---|---|---|---|---|---|---|---|---|---|---|
| AllSample | 27.75% | 14.00% | -56.38% | 9.88% | 16.13% | 20.38% | 24.25% | 27.75% | 31.25% | 35.00% | 39.50% | 45.75% | 93.88% |
| CashCoIn | 25.25% | 15.88% | -41.63% | 5.50% | 12.00% | 16.75% | 20.75% | 24.75% | 28.50% | 32.50% | 37.88% | 45.25% | 93.75% |
| CashOut | 30.63% | 15.00% | -33.75% | 11.63% | 18.25% | 22.88% | 26.75% | 30.50% | 34.13% | 38.13% | 43.13% | 49.88% | 93.75% |
| ExNLargeCoInAct | 23.88% | 15.00% | -38.13% | 5.13% | 11.38% | 15.88% | 19.50% | 23.38% | 27.00% | 31.13% | 35.88% | 43.13% | 93.50% |
| ExNLargeCoOutAct | 31.13% | 15.13% | -31.75% | 12.38% | 19.00% | 23.38% | 27.38% | 31.25% | 34.63% | 38.88% | 43.50% | 50.38% | 93.63% |
| MedSmallCoInAct | 27.50% | 15.00% | -41.00% | 8.50% | 14.75% | 19.38% | 23.38% | 27.00% | 30.75% | 34.88% | 39.25% | 46.25% | 92.88% |
| MedSmallCoOutAct | 28.88% | 14.38% | -35.63% | 10.63% | 16.88% | 21.25% | 25.00% | 28.75% | 32.63% | 36.38% | 40.88% | 47.50% | 91.63% |

Figure 2: Statistics on the correlation coefficient of returns of stocks associated with the full sample and capital flow similarity graph, and data deadline is December 31, 2021.

In the six correlation charts, the distribution of correlation coefficients among stocks associated with capital outflows, large-scale capital outflows, and small-to-medium capital outflows displays a noticeable increase compared to the overall market. The average and various quantiles of the correlation coefficients for the three types of capital outflow samples are 1%-5% higher than those of the entire sample.

## 4.3 EFFECTIVENESS ANALYSIS OF THE PROPOSED GRAPH FACTORS

To evaluate the performance of our proposed factors, we displays the information coefficient (IC) and ranked IC (rankIC) of the training schemes for each graph on the constituent stocks of the China Composite Index and their future 10-day returns in Figure 3. The average IC of the factors in the fund-related flow similarity matrix, generated by overlaying a single graph, ranges from 9.38% to 9.52%. This is 0.1% to 0.2% higher than the output of the original factor model.

| | Graph Factors | IC | IC means | IC std | Rank IC | IC_IR | IC Winning Rate |
|---|---|---|---|---|---|---|---|
| China Securities Index | Ens_LRI | 9.67% | 10.69% | 11.85% | 10.75% | 0.9043 | 80.87% |
| | Ens_TRI | 9.60% | 10.48% | 11.67% | 10.56% | 0.9158 | 81.40% |
| | Ens_TRU | 9.68% | 10.71% | 11.84% | 10.80% | 0.9032 | 81.17% |
| | CashCoIn | 9.49% | 10.45% | 11.57% | 10.49% | 0.9078 | 81.33% |
| | CashOut | 9.42% | 10.59% | 11.67% | 10.59% | 0.8898 | 80.34% |
| | ExNLargeCoInAct | 9.52% | 10.40% | 11.64% | 10.47% | 0.9158 | 81.33% |
| | ExNLargeCoOutAct | 9.46% | 10.53% | 11.69% | 10.59% | 0.8984 | 80.56% |
| | MedSmallCoInAct | 9.41% | 10.49% | 11.50% | 10.59% | 0.8970 | 81.25% |
| | MedSmallCoOutAct | 9.38% | 10.57% | 11.61% | 10.62% | 0.8867 | 80.41% |
| | RawModel | 9.35% | 10.41% | 11.57% | 10.51% | 0.8985 | 80.34% |

Figure 3: Superimpose the model IC and rankIC obtained by the joint training scheme of each map in the whole A share market. The data deadline is December 31, 2022.

With the exception of the small and medium-sized order common outflow graph, the fund flow similarity extraction factor demonstrates varying increments in IC and rankIC. The average IC output of the model with multiple graphs superimposed on the factors of the fund flow relationship matrix ranges from 9.60% to 9.68%, about 0.3% higher than the output of the original factor model.

Moreover,we illustrate the cumulative IC increment curves of each scheme relative to the original model in the Fig. 4. In the first sub-figure, We can see that the single graph scheme exhibits relatively stable IC increments before 2020. However, the cumulative IC increments experience collective failure and differentiation among different schemes after 2020. The CashOut and the ExNLargeCoOurAct consistently enhance the performance of our model throughout the entire testing cycle. The ExNlargeCoInAct performs best before July 2022, but experiences a decline afterward with unstable enhancements during the cycle.

However, the multi-graph factors, LRI and TRU in the second sub-figure, demonstrate a significant continuous improvement in IC increments. Compared to single graph, the multi-graph factors exhibit stronger and sustained stability of increments.

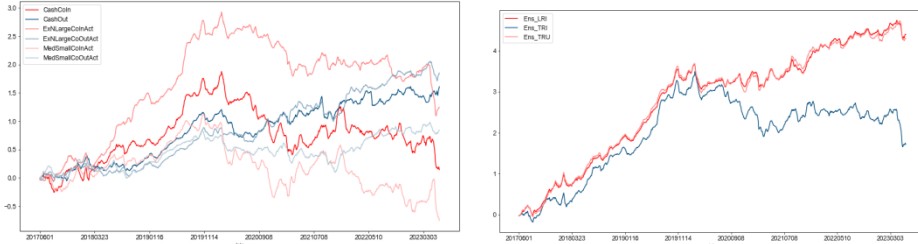

Figure 4: The IC increment of each fund-related graph factors and derived factors which are accumulated during training.

## 4.4 EFFECTIVENESS ANALYSIS OF THE SUPERIMPOSED FUND-RELATED GRAPHS MODEL ON CONSTITUENT STOCKS OF A-SHARE MARKET

To further evlauate the effects of our proposed factors across the constituent stocks of different indices in A-share market, more models are trained with superimposed fund-related graphs to check different degrees of improvement in IC and rankIC. The results are shown in the Fig 5. We can see that the volatility of IC and rankIC also increases accordingly. In terms of absolute increment, there is a larger IC increment within the constituent stocks of the CSI 300 Index, while the increments gradually weaken for the CSI 800, CSI 500, and CSI 1000 indices. The differences in the relative strength of enhancement effects among different types of proposed graph factors may stem from various reasons, such as the dominant drivers of fund flow similarity for stocks with larger market capitalization, which are more deterministic. Therefore, there is a closer relationship between stock price movements and fund flow similarity. On the other hand, stocks with smaller market capitalization exhibit more randomness in their fund flow similarity.

| | Graph Factors | IC | IC means | IC std | Rank IC | IC_IR | IC Winning Rate |
|---|---|---|---|---|---|---|---|
| CSI1000 | Ens_LRI | 7.01% | 13.23% | 8.02% | 12.83% | 0.5299 | 69.82% |
| | Ens_TRI | 6.99% | 12.96% | 7.89% | 12.63% | 0.5394 | 70.43% |
| | Ens_TRU | 7.04% | 13.26% | 8.05% | 12.86% | 0.5309 | 70.20% |
| | CashCoIn | 6.83% | 12.80% | 7.76% | 12.40% | 0.5334 | 70.73% |
| | CashOut | 6.60% | 12.90% | 7.72% | 12.41% | 0.5120 | 69.66% |
| | ExNLargeCoInAct | 6.88% | 12.78% | 7.85% | 12.38% | 0.5380 | 70.05% |
| | ExNLargeCoOutAct | 6.71% | 12.83% | 7.84% | 12.35% | 0.5232 | 69.82% |
| | MedSmallCoInAct | 6.72% | 12.72% | 7.71% | 12.27% | 0.5282 | 69.74% |
| | MedSmallCoOutAct | 6.58% | 12.82% | 7.73% | 12.35% | 0.5129 | 69.89% |
| | RawModel | 6.55% | 12.55% | 7.71% | 12.10% | 0.5220 | 68.75% |
| CSI500 | Ens_LRI | 6.28% | 15.82% | 6.86% | 15.72% | 0.3973 | 66.46% |
| | Ens_TRI | 6.35% | 15.45% | 6.85% | 15.32% | 0.4107 | 66.46% |
| | Ens_TRU | 6.34% | 15.79% | 6.86% | 15.70% | 0.4016 | 66.31% |
| | CashCoIn | 6.13% | 15.18% | 6.64% | 14.98% | 0.4039 | 66.23% |
| | CashOut | 5.86% | 15.35% | 6.50% | 15.22% | 0.3820 | 65.47% |
| | ExNLargeCoInAct | 6.18% | 15.16% | 6.74% | 14.99% | 0.4076 | 67.00% |
| | ExNLargeCoOutAct | 5.80% | 15.18% | 6.46% | 15.05% | 0.3820 | 65.09% |
| | MedSmallCoInAct | 6.01% | 14.81% | 6.51% | 14.63% | 0.4061 | 65.85% |
| | MedSmallCoOutAct | 5.73% | 15.06% | 6.39% | 14.89% | 0.3807 | 65.40% |
| | RawModel | 5.69% | 14.72% | 6.36% | 14.54% | 0.3864 | 65.47% |

| | Graph Factors | IC | IC means | IC std | Rank IC | IC_IR | IC Winning Rate |
|---|---|---|---|---|---|---|---|
| CSI800 | Ens_LRI | 9.20% | 11.23% | 11.02% | 11.20% | 0.8196 | 78.58% |
| | Ens_TRI | 9.07% | 11.08% | 10.79% | 11.03% | 0.8190 | 78.43% |
| | Ens_TRU | 9.20% | 11.28% | 11.02% | 11.28% | 0.8157 | 78.43% |
| | CashCoIn | 8.96% | 11.07% | 10.66% | 10.97% | 0.8096 | 78.20% |
| | CashOut | 8.97% | 11.09% | 10.85% | 11.01% | 0.8092 | 78.73% |
| | ExNLargeCoInAct | 9.06% | 10.93% | 10.82% | 10.91% | 0.8295 | 78.20% |
| | ExNLargeCoOutAct | 9.06% | 11.08% | 10.90% | 11.03% | 0.8182 | 78.43% |
| | MedSmallCoInAct | 8.91% | 11.10% | 10.69% | 11.10% | 0.8029 | 78.20% |
| | MedSmallCoOutAct | 9.01% | 11.14% | 10.82% | 11.05% | 0.8088 | 78.81% |
| | RawModel | 8.94% | 10.95% | 10.76% | 10.94% | 0.8168 | 78.43% |
| CSI300 | Ens_LRI | 7.52% | 13.28% | 8.90% | 12.54% | 0.5660 | 70.88% |
| | Ens_TRI | 7.43% | 13.03% | 8.69% | 12.36% | 0.5703 | 71.04% |
| | Ens_TRU | 7.51% | 13.34% | 8.88% | 12.60% | 0.5627 | 71.04% |
| | CashCoIn | 7.29% | 12.93% | 8.59% | 12.19% | 0.5637 | 70.88% |
| | CashOut | 7.12% | 12.97% | 8.60% | 12.16% | 0.5493 | 70.35% |
| | ExNLargeCoInAct | 7.39% | 12.87% | 8.69% | 12.15% | 0.5739 | 70.73% |
| | ExNLargeCoOutAct | 7.36% | 13.03% | 8.86% | 12.20% | 0.5646 | 70.27% |
| | MedSmallCoInAct | 7.15% | 12.99% | 8.51% | 12.22% | 0.5504 | 69.82% |
| | MedSmallCoOutAct | 7.18% | 13.04% | 8.73% | 12.24% | 0.5502 | 70.81% |
| | RawModel | 7.17% | 12.75% | 8.69% | 12.02% | 0.5618 | 70.20% |

Figure 5: The performance of proposed fund-related factors in four different constituent stocks of the A-share market, including CSI1000, CSI800, CSI500, and CSI300.

## 5 CONCLUSION

In this paper, our research focus on the effects of the capital flow in the A-share market. Six fund-related graph factors and three derived factors are proposed for analysis. Combining with some high-efficent factors, a new representation factors are formed. Results show that the new fund-related factors achieve better performance proving that the proposed fund-related graph factors can obtain maginal effectiveness in quantitative strategy.

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
