# OpenReview forum: "FUND-RELATED GRAPH REPRESENTATION FOR MARGINAL EFFECTIVENESS IN MULTI-FACTORS QUANTITATIVE STRATEGY"
_ICLR.cc/2024/Conference — Submitted to ICLR 2024_

### Official Review · Reviewer_Ey9j · 2023-10-26

**Soundness:** 2 fair
**Presentation:** 2 fair
**Contribution:** 2 fair
**Rating:** 5
**Confidence:** 3

**Summary:**

This study explores enhancing quantitative trading using multi-factor machine learning and relational graph data. Researchers developed new factors using six capital flow similarity graphs to understand stock fund flows. When integrated into an XGBoost model, these graph-based factors outperformed traditional models in several key market performance metrics, highlighting the potential of graph data in improving quantitative trading strategies.

**Strengths:**

1. The authors proposed an interesting and practical solution to improve factor quality by incorporating relational information.
2. The paper provides intuitive explanations and visualizations of the constructed graphs and the effects of using them for factor derivation. The concepts are clearly explained and easy to follow.
3. The paper conducts extensive experiments on real-world A-share market data to evaluate the proposed methods. The results demonstrate improved performance in terms of information coefficient, excess returns, etc when using the derived graph factors.

**Weaknesses:**

1. The paper does not provide sufficient details on the baseline factor library used. More information on the existing factors would help better contextualize the incremental value added by the proposed graph factors.
2. The authors only compare the method to a simple baseline where no relational information is needed. The authors should evaluate the effects (1) The method to incorporate relational information (2) How to obtain relational information. The authors should include baselines to extract relational information from other sources and literature.
3. The paper considers only one machine learning model (XGBoost) for experimentation. Evaluating with other models like neural networks could demonstrate the wider applicability of the proposed techniques.
4. The presentation of the paper can be improved. The legends in Figures 4 is too small and blur to be read. It is not clear what the colors represent in tables. There are also several missing space before citations.

**Questions:**

Please see weaknesses

---

### Official Review · Reviewer_FWmh · 2023-10-30

**Soundness:** 2 fair
**Presentation:** 3 good
**Contribution:** 2 fair
**Rating:** 3
**Confidence:** 4

**Summary:**

This paper examines the process of constructing new factors from the relational graph data. It constructs six capital flow similarity plots from the frequency of net fund inflows or outflows between stocks over the same period. Furthermore, it forms three composite multigraphs by combining three of the six primitive graphs to leverage capital flow similarity. The authors empirically verify the incremental value of these newly proposed graphs by employing a multifactor XGBoost model, which incorporates a representation of the fund correlation graph into the baseline multifactor model.

**Strengths:**

1.	This paper constructs six basic graphs from fund-related data to exploit the representation of capital flow similarities.
2.	The experimental results prove that the addition of capital flow similarity brings an enhanced effect on the revenue prediction.

**Weaknesses:**

1.	The contribution of this paper appears to be limited, as it primarily focuses on suggesting the construction of graphs to improve the performance of the multifactor XGBoost model, without introducing substantial innovations to the model itself.
2.	The comparative experiments in this paper lack comprehensiveness as they solely compare the XGBoost model with derived features that incorporate graphs to the XGBoost model with original features. Furthermore, there is an absence of comparative experiments involving other models. For example:[1],[2]。
3.	The experiment in this paper lacks regional validity because it was validated only on the Chinese stock dataset and not on datasets from other regions, e.g., the U.S., and Europe.
4.	This paper does not offer code or datasets, thereby it lacks reproducibility.

[1]Shihao Gu, Bryan Kelly, and Dacheng Xiu. Autoencoder asset pricing models. Journal of Econometrics, 222(1):429–450, 2021.

[2] Yitong Duan, Lei Wang, Qizhong Zhang, and Jian Li. Factorvae: A probabilistic dynamic factor model based on variational autoencoder for predicting cross-sectional stock returns. In Proceedings of the AAAI Conference on Artificial Intelligence, volume 36, pp. 4468–4476, 2022.

**Questions:**

In Section 3.2, customer service orders are classified into four categories based on transaction amounts. For hyperparameter experiments, it is possible to explore different interval ranges for transaction amounts. Additionally, hyperparameter experiments can be conducted to vary the proportions of strong connections retained during the composition process, allowing for adjustments in the retained proportions.

---

### Official Review · Reviewer_D9Ht · 2023-11-01

**Soundness:** 3 good
**Presentation:** 2 fair
**Contribution:** 2 fair
**Rating:** 3
**Confidence:** 2

**Summary:**

The authors propose and explore several ways to construct representations or features from financial relational graph data. The authors then use these representations together with an XGBoost model to predict the return of stocks. The experimental results show that the proposed method's predictions have high agreement with the true value.

**Strengths:**

- Proposed method is reasonable and leverages in domain knowledge

**Weaknesses:**

- Limited impact. The proposed graph construction seems to be very specific to the task at hand, which limits the impact of this work.
- No comparison to alternative approaches.

**Questions:**

It is unclear what the impact of the graphs is. What if we just train an XGBoost model with all the available features?

---

### Meta-Review · Area_Chair_1AUk · 2023-12-04

**Metareview:**

The paper proposed to construct representations or features from relational graph data for prediction. The reviewers share concerns in

- Insufficient comparison with baseline in experiments
- Limitation in novelty and scope.

**Justification For Why Not Higher Score:**

The reviewers shared concerns about experiments and the contribution of the submission.

**Justification For Why Not Lower Score:**

N/A

---

### Decision · Program_Chairs · 2024-01-16

Reject